# Effect of prosocial public health messages for population behaviour change in relation to respiratory infections: a systematic review protocol

Aikaterini Grimani [1,2] Chris Bonell [1,3] Susan Michie,[1,4] Vivi Antonopoulou,[1,4] Michael P Kelly,[1,5] Ivo Vlaev [1,2]

¹NIHR Policy Research Unit in Behavioural Science, Newcastle University, Newcastle upon Tyne, UK
²Warwick Business School, University of Warwick, Coventry, UK
³Public Health and Policy, London School of Hygiene & Tropical Medicine, London, UK
⁴UCL Psychology and Language Sciences, University College London, London, UK
⁵School of Clinical Medicine, University of Cambridge, Cambridge, UK

**Correspondence to**
Dr Aikaterini Grimani;
aikaterini.grimani@wbs.ac.uk

## ABSTRACT

**Introduction** The COVID-19 pandemic represents a major societal challenge that requires large-scale behaviour change, widespread collective action and cooperation to reduce viral transmission. Existing literature indicates that several messaging approaches may be effective, including emphasising the benefits to the recipient, aligning with the recipient's moral values and focusing on protecting others. Current research suggests that prosocial public health messages that highlight behaviours linked to societal benefits (eg, protecting 'each other'), rather than focusing on behaviours that protect oneself (eg, protecting 'yourself'), may be a more effective method for communicating strategies related to infectious disease. To investigate this we will conduct a systematic review that will identify what messages and behaviour change techniques have the potential to optimise the effect on population behaviour in relation to reducing transmission of respiratory infections.

**Methods and analysis** A systematic literature search of published and unpublished studies (including grey literature) in electronic databases will be conducted to identify those that meet our inclusion criteria. The search will be run in four electronic databases: MEDLINE, EMBASE, PsycINFO and Scopus. We will also conduct supplementary searches in databases of 'grey' literature such as PsycEXTRA, Social Science Research Network and OSF PREPRINTS, and use the Google Scholar search engine. A systematic approach to searching, screening, reviewing and data extraction will be applied based on Preferred Reporting Items for Systematic Reviews and Meta-Analyses. Titles, abstracts and full texts for eligibility will be examined independently by researchers. The quality of the included studies will be assessed using the Cochrane Risk of Bias Tool and the Risk of Bias in Non-randomized Studies-of Interventions tool. Disagreements will be resolved by a consensus procedure.

**Ethics and dissemination** This protocol has been registered with PROSPERO. No ethical approval is required, as there will be no collection of primary data. The synthesised findings will be disseminated through peer-reviewed publication.

**PROSPERO registration number** CRD42020198874.

## Strengths and limitations of this study

► The analysis, which will be carried out using the Behaviour Change Wheel framework, will identify the key behaviours and, importantly, drivers for behaviours that may be amenable to change.
► The review will include narratively synthesised sections such as target behaviour, intervention, context, mechanisms of action, outcome and effect.
► The inclusion of grey literature will broaden this study in terms of included information.
► A limitation of this review will be the exclusion of papers reported in languages other than English.

## INTRODUCTION

The COVID-19 pandemic represents a massive global health crisis which requires the rapid adoption of population protective and physical social distancing behaviours.[1] To prevent infectious disease transmission in a pandemic, effective strategies are required to communicate important health messages in a concise and meaningful way that makes it easy for citizens to change behaviour, and avoiding community ambivalence and panic.[2 3] Several messaging approaches may be effective, including emphasising the benefits to the recipient, focusing on protecting others (eg, 'wash your hands to protect your parents and grandparents'; 'coughing or sneezing into your elbow to protect other people'), aligning with the recipient's moral values, appealing to social consensus or scientific norms and/or highlighting social group approval.[4] Existing literature reveals the importance of communications and public messaging during pandemic outbreaks on the effective implementation of, and adherence to, face mask and other non-pharmaceutical interventions.[5] Inconsistent information from various sources prompts individuals to question the credibility of available information, sometimes resulting in fear and denial of the pandemic. Relevant studies have found that when the communication efforts are seen as

unreliable, inconsistent, sensationalist or unduly alarmist, doubts are likely to be generated. These doubts are likely to influence public behavioural responses to an emerging respiratory infection outbreak and may also lead to people disregarding future advice.[6–8] Considering that the main function of wearing a mask is to reduce disease transmission, wearers' behaviour could be depicted as altruistic. This protective behaviour regarding wearing a mask could promote social responsibility and solidarity against a common threat. Campaigns in several countries rapidly reshaped societal norms around the acceptability of wearing a mask in public. Social media were used to share messaging about making them at home or to show celebrity support for the campaign, creating a movement, which prompted others to imitate this behaviour and follow the example. Slogans such as 'my mask protects you, your mask protects me' seem to be considered attractive on a set of social norms.[9 10]

Enlisting trusted voices has been shown to make public health messages more effective in changing behaviour during epidemics.[4 11 12] Abu-Akel et al[13] suggested that different spokespersons may be needed for the younger and the older populations, and for rural and urban populations. They found that a government official was more effective than a celebrity actor, particularly in response to compliance with social distancing measures, and was considered substantially stronger by older respondents despite them having lower risk perception compared with younger respondents. Current findings also suggest that the thought of infecting vulnerable people or large numbers of people can motivate social distancing.[11 14–17] It may therefore be useful to identify which public health messages work best on which populations, in order to generate policy support and to ensure individuals' actions help reduce infectious disease transmission.

A group of behavioural and social scientists, who have shared their advice with UK government on COVID-19, have collaborated to develop a series of principles to inform interventions to promote whole population adherence to social distancing. The experts resulted to 11 key principles based on their expertise and the knowledge of existing theory and evidence.[18] A promising key principle, which promotes care for others rather than individual self-interest, is based on social identity,[19] social influence[20] and moral behaviour.[21] These 'Protect each other' messages highlight the benefits of protective behaviours for the group and its most vulnerable members, including our loved ones, with evidence of benefits in the COVID-19[22] and other health contexts.[23] For example, young people are responsible for the majority of transmissions of influenza,[24] but the majority of deaths happen over 65 years of age.[25] Such messages are usually enhanced by concrete examples, powerful images and the actual voices of those we need to protect, linked to clear, specific advice on how to implement desired behaviours.[7 18 26] In order to persuade those who are willing to help others to act in the collective interest, such messages sometimes use images and accounts of widespread population adherence.[18 27]

In addition, such messages are expected to be efficacious for those who consider themselves at low risk of severe consequences from COVID-19 infection due to optimism bias.[18]

According to the group of experts, the other 10 key principles are:
► Messages should come from representative and trusted personages who can influence and urge diverse groups to help each other.
► Messages should reflect group culture and behaviour using social norms.
► Messages should give clear, specific and calm advice, enabling people to plan how to commit to protective behaviours and review these plans regularly.
► Use a theory of change of how campaign activities aim to generate behavioural impacts.
► Avoid messages based on fear/disgust in relation to other people.
► Avoid authoritarian messages, as evidence shows that individuals differ markedly in their receptiveness to what may be seen as authoritarian moral messages.
► Messages which promote reward, incentives and enablement tend to be more effective influences on desired behaviours.
► Messages should be communicated via professionally designed and appealing mass and social media campaigns with trusted spokespeople.
► There is a need for clear and specific guidance on the protective behaviours the general population should adopt.
► The interventions should be codesigned, piloted and evaluated using a range of methods.[18]

Webster et al's[28] rapid review of the evidence on how to improve adherence with quarantine supports some of the principles mentioned above. These include: providing a timely, clear rationale for quarantine, providing clear information about quarantine protocol, emphasising social norms to encourage the behaviour as altruistic, highlighting the benefit that engaging in quarantine will have for public health, emphasising its importance (in particular to those at heightened risk of the disease), and ensuring sufficient supplies are provided, providing assistance for those financially affected by quarantine.

The existing literature illustrates that prosocial public health messages that highlight behaviours related to societal and communal benefits (eg, protect each other), rather than focusing on behaviours that only benefit the self (eg, protect yourself), may be an especially effective method[29–31] for communicating public health recommendations related to infectious diseases.[7 32 33] Prosocial behaviour is defined as behaviour that benefits others, whether or not it involves an overall cost to self, and includes a variety of important social behaviours such as helping, sharing and cooperation.[34 35] The term 'others' refers to specific individuals or groups of people, and in particular loved ones, vulnerable members (with weakened immune systems such as elderly and chronically ill), healthcare professionals, coworkers, workers in retail and

public services, members of the public and, therefore, the society as a whole.

Given the above, we will conduct a systematic review that will identify what messages and behaviour change techniques (BCT) have the potential to optimise the effect on population behaviour in relation to reducing transmission of respiratory infections. The evidence will be gathered from infectious diseases field across a range of behaviours (such as hand hygiene, social distancing, face masks, touching face, catching coughs/sneezes, testing, reporting contacts, self-isolating if infected), and with a specific focus on populations where a protect-others-message is important, that is, those already infected and those at low risk of serious consequences of infection (young people in influenza or coronavirus epidemics). Such insights are important for government health agencies which will benefit from the evidence regarding effective communication strategies as a promising instrument to prevent infectious disease transmission.

## Objective

The aim of this research is to conduct a systematic review of effective communication strategies for population behaviour change in relation to infectious diseases and to synthesise the evidence on supporting or/and refining the 'protect each other' principle.

Research questions:

▶ Are messages focusing on protecting others effective in changing a defined list of behavioural outcomes compared with other messages/controls?
▶ What behaviours (eg, social distancing, hand washing, face touching, using hygiene products, and so on) do messages about protecting others appear to affect positively?
▶ What populations do messages about protecting others appear to affect positively?

## METHODS AND ANALYSIS

The systematic review will be performed following the Preferred Reporting Items for Systematic Reviews and Meta-Analyses (PRISMA) statement.[36 37] The protocol was registered in the International Prospective Register of Systematic Reviews (PROSPERO).[38]

A comprehensive literature search of published and unpublished studies ('grey literature') in electronic databases will be conducted. The following electronic databases will be searched from inception to October 2020: MEDLINE, EMBASE.com, PsycINFO and Scopus. For unpublished studies we will conduct searches in databases of 'grey' literature such as PsycEXTRA, Social Science Research Network and OSF PREPRINTS database which includes BioHackrXiv, Cogprints, MediArXiv, SocArXiv, PsyArXiv and RePEc. We will also conduct supplementary searches in Google Scholar, hand search relevant journals and conduct backward and forward citation searching of included studies and relevant reviews.

A search strategy (online supplemental table 1) following Population, Intervention, Comparison, Outcome and Study design (PICOS) will be adapted, including Medical Subject Headings terms and relevant keywords.[39 40] The PICOS process will facilitate an evidence-based approach to literature searching and help to rapidly and accurately locate the best available scientific information, limiting unnecessary searching.[40]

## Eligibility criteria

Only studies written in the English language will be considered. The specific inclusion criteria of the study will be the following:

▶ Types of studies to be included: We will include published and unpublished (grey literature) randomised controlled trials, quasiexperimental studies or time series analyses.
▶ Population: We will include the general public including population with or without vulnerabilities and/or people who are/are not already infected. There will be no restrictions placed on geographical region.
▶ Interventions: We will be looking for interventions focused on communication via mass media, social media or print media (such as leaflets and posters) or health professional advices via consultation where messages about protecting others are included.
▶ Comparisons: Relevant comparisons will include no message or active control with messages focused on self-protection or messages that contain no motivational content.
▶ Primary outcomes: We will include any behaviour relevant to reducing transmission of respiratory infections (eg, socialising, hand hygiene, social distancing, face masks, touching face, catching coughs/sneezes, testing, reporting contacts, isolating if infected).

## Patient and public involvement

A core component of Policy Research Unit (PRU) policy is to maximise public involvement in all our research activities.[41] Thus, the PRU patient and public involvement panel has commented on the protocol and will receive regular updates on the review and comment on outputs.

## Selection of studies

All titles and abstracts retrieved by electronic searching will be downloaded to the reference manager database EndNote and uploaded to Covidence, one of Cochrane's recommended tools. This web-based software platform has been designed to support more efficient management of systematic reviews and can be used from the beginning of title/abstract screening through the beginning of meta-analysis.[42] Duplicates will be removed and double screening will be done on a proportion of the retrieved citations until the appropriate agreement (>95%) is achieved. The remaining studies will be screened by one reviewer. The abstracts will be included if they meet the inclusion criteria. The same procedure will be applied to determine the eligibility of studies on the basis of a review of the full texts. Differences in judgement will

be resolved through discussion and inclusion of a third researcher doing the rating, if required. The selection process will be recorded and the PRISMA flow diagram will be completed.[36]

## Data extraction

The process of data extraction will involve one or two reviewers who will generate a data extraction form. The theoretical underpinnings of intervention content will also be considered. The data extraction form, which will be reviewed and refined by the reviewers, will include variables as follows:

► The communication message.
► Characteristics of the recipient(s) of the communication (protect-others-message).
► Characteristics of the 'others' who will be protected due to the message.
► The manner in which the appeal to protect others is made.
► Intervention features including:
  MINDSPACE checklist of behavioural economic techniques: Messenger, Incentives, Norms, Defaults, Salience, Priming, Affect, Commitment and Ego.[43 44]
  BCTs (eg, observable and replicable components designed to change behaviour),[45] intervention functions (education, persuasion, incentivisation, coercion, training, restriction, environmental restructuring, modelling, enablement) and policy categories (communication/marketing, guidelines, fiscal, regulation, legislation, environmental/social planning, service provision), which are all part of the Behaviour Change Wheel framework.[46]
  Primary outcome(s): Any behaviour relevant to reducing transmission of respiratory infections. Where more than one reported outcome is provided, we will use the Grades of Recommendation, Assessment, Development and Evaluation approach to assess the certainty of evidence (separated into those that are critical (any behaviour relevant to reducing transmission of respiratory infections) and not important (other outcomes)).[47] Outcomes measured at multiple time points will be categorised as follows: immediate (within 2 weeks of the intervention delivery), short-term (2–13 weeks after intervention delivery), medium-term (14–50 weeks after intervention delivery) and long-term effects (51 or more weeks after intervention delivery). We will present multiple time points only for critical outcomes. The downgrading of the quality of a body of evidence for a specific outcome will be based on five factors: limitations of study; indirectness of evidence; inconsistency of results; precision of results; and publication bias. Results.

## Quality assessment

An assessment of the methodological quality of included studies will be conducted using the Cochrane Collaboration Risk of Bias Tool (RoB 2)[48] and the

Risk of Bias in Non-randomized Studies-of-Interventions (ROBINS-I).[49 50] RoB 2 is structured into five bias domains (bias arising from the randomisation process; bias due to deviations from intended interventions; bias due to missing outcome data; bias in measurement of the outcome; bias in selection of the reported result). Within each domain, the assessment comprises: a series of signalling questions; a judgement about risk of bias for the domain, facilitated by an algorithm that maps responses to signalling questions to a proposed judgement; free text boxes to justify responses to the signalling questions and risk of bias judgements; optional free text boxes to predict (and explain) the likely direction of bias. The response options are 'yes', 'probably yes', 'probably no', 'no' and 'no information'. The risk of bias judgements for each domain are 'low risk of bias', 'some concerns' or 'high risk of bias'. According to the Cochrane Collaboration's recommendations, the studies, whose all domains will be rated low risk, will be judged as low risk of bias, while the studies with one or more concerns will be judged to raise some concerns. Furthermore, the studies, where one or more domains will be rated high risk, will be judged to be at high risk of bias.[51]

ROBINS-I is mainly distinct from RoB 2 because of the randomisation. The first two domains, covering confounding and selection of participants into the study, address issues before the start of the interventions that are to be compared ('baseline'). The third domain addresses classification of the interventions themselves. The other four domains address issues after the start of interventions: biases due to deviations from intended interventions, missing data, measurement of outcomes and selection of the reported result. The categories for risk of bias judgements are 'Low risk', 'Moderate risk', 'Serious risk' and 'Critical risk' of bias. If none of the answers to the signalling questions for a domain suggests a potential problem then risk of bias for the domain can be judged to be low. Otherwise, potential for bias exists.[49] Two review authors will independently evaluate the methodological quality of each included study using the assessment tools. Discrepancies will be resolved through a consensus procedure.

## Data analysis and synthesis

We will compare the communication strategies where messages focus on protecting others against an alternative condition. Randomised trials and quasiexperimental studies will be analysed to develop a taxonomy of interventions where possible describing these in terms of BCTs and, when appropriate, meta-analytically. In particular, the review will include narratively synthesised sections such as:

► Target behaviour: What is the desired response? This step involves clearly defining the behavioural response required by the recipient(s) of the communication.
► Intervention: A product, service, activity or structural change, intended to achieve behaviour change. BCTs will be double coded by trained coders using the

Behaviour Change Technique Taxonomy v1[45] using a BCT extraction form. BCTs will be coded separately for intervention and control groups. The MINDSPACE checklist[43] will also be used as it adds techniques predominantly used in the behavioural economics literature. The reliability of coding of BCTs and MIND-SPACE checklist will be assessed using the prevalence and bias-adjusted kappa (PABAK) statistic.[52] PABAK will be used because it adjusts for shared bias in the coders' use of categories and high prevalence of negative agreement (eg, when both coders agree that codes are absent). We will calculate pooled effect sizes within each pairwise comparison (eg, intervention type vs control), accounting for the extent of heterogeneity among the studies. If an indication of substantial heterogeneity is determined that cannot be explained through meta-regressions, then we will not produce a pooled estimate and will present only the narrative summary. If we consider that we have high levels of unexplained statistical heterogeneity in any of our study groupings, we will investigate this further using subgroup and sensitivity analyses. As is appropriate, a random-effects meta-analysis with the extracted BCTs will be conducted. The meta-analysis will be conducted when a group of studies will be sufficiently homogeneous in terms of subjects involved, interventions and outcomes to provide a meaningful summary. Although a meta-analysis can be conducted with a minimum of two studies, Valentine *et al*[53] suggest that the combination of very few studies with heterogeneous characteristics makes any kind of synthesis untenable in most cases, while parameter estimation (eg, the random-effects variance component) will likely be poor, rendering conclusions that are highly uncertain.

► Context: Attributes of the target population (characteristics of the individuals, groups, subpopulations or populations) and the intervention setting (eg, social and physical environment).

► Mechanisms of action (MoA): Processes by which intervention influences the target behaviour (moderator factors).[54] The 26 MoA, which are defined as the processes through which a BCT affects behaviour, are taken from the 12 most frequently occurring mechanisms derived from a set of 83 behaviour change theories[46] and the 14 theoretical domains as described in the Theoretical Domains Framework (TDF: knowledge; skills; social/professional role and identity; beliefs about capabilities; optimism; beliefs about consequences; reinforcement; intentions; goals; memory, attention and decision processes; environmental context and resources; social influences; emotions; and behavioural regulation).[55] The TDF, which is developed by synthesising 33 psychological theories aimed at behaviour change, enables interventionists to better understand the psychological mechanisms of change.[46 56]

► Outcome: The type of the target behaviour in the given scenario (eg, how and when it is assessed).

► Effect: An estimate of the comparison between the outcomes in the evaluated scenarios.

We will also focus on implications for intervention research. We will assess whether interventions addressing the various outcomes, or some subsets of these, appear to have mediating factors or mechanisms.

## ETHICS AND DISSEMINATION

Ethics approval is not required for this knowledge synthesis. This protocol had been registered with the PROSPERO. The synthesised findings will be disseminated through peer-reviewed publication. This knowledge synthesis can serve as a guide for effective communication methods used to promote adherence to behaviours that prevent infectious disease transmission during pandemics (such as COVID-19), epidemics or endemics for National Health System and Department of Health and Social Care stakeholders and other stakeholders. Reaching people with a message is one thing, influencing and changing their behaviour is quite another. Our behavioural science review will provide tried and tested methods that help us design communication strategies that more effectively influence population behaviour relevant to reducing transmission of respiratory infections.

**Contributors** AG is the corresponding author and the primary author of the study, conceived the study, contributed to the development of the search strategy, developed the inclusion and exclusion criteria and data extraction criteria and was involved in the conceptualisation of the research questions. CB conceived the study, contributed to the development of the selection criteria and data extraction criteria, was involved in the conceptualisation of the research questions and provided written feedback on the manuscript. SM conceived the study and provided written feedback on the manuscript. VA contributed to the development of the search strategy and provided written feedback on the manuscript. MPK contributed to the development of the selection criteria and data extraction criteria and provided written feedback on the manuscript. IV conceived the study, contributed to the development of the selection criteria and data extraction criteria, was involved in the conceptualisation of the research questions and revised the manuscript critically and contributed to it intellectually. All authors have read and approved the final version of the manuscript.

**Funding** This study/project is funded by the National Institute for Health Research (NIHR) (Policy Research Unit in Behavioural Science (project reference PR-PRU-1217-20501)).

**Disclaimer** The views expressed are those of the author(s) and not necessarily those of the NIHR or the Department of Health and Social Care.

**Competing interests** None declared.

**Patient consent for publication** Not required.

**Provenance and peer review** Not commissioned; externally peer reviewed.

## ORCID iDs

Aikaterini Grimani http://orcid.org/0000-0002-2076-6199
Chris Bonell http://orcid.org/0000-0002-6253-6498
Ivo Vlaev http://orcid.org/0000-0002-3218-0144

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
