## [Reviewer comments · BMJ Open]

ARTICLE DETAILS

TITLE (PROVISIONAL)	The effect of prosocial public health messages for population behaviour change in relation to respiratory infections: A Systematic Review protocol
AUTHORS	Grimani, Aikaterini; Bonell, Chris; Michie, Susan; Antonopoulou, Vivi; Kelly, Michael; Vlaev, Ivo

VERSION 1 – REVIEW

REVIEWER	Pete Lunn Economic and Social Research Institute (ESRI) & Trinity College Dublin, Ireland
REVIEW RETURNED	23-Oct-2020

GENERAL COMMENTS	This is a strong idea, which is expertly described and will make an important contribution. I have some minor comments: 1) An important limitation is omitted, which is the difficulty associated with trying to conduct a thorough review when so many studies are likely to be conducted but not written-up and peer-reviewed. Greater thought needs to be given to how this issue will be handled when weighting evidence. 2) I think the eligibility criteria need to be tweaked. A control group of 'no message' or 'self-protection' excludes control messages that contain no motivational content (e.g. purely informational messages). I think this exclusion would be a mistake. 3) I have a concern that current behaviour change theories are at their weakest when it comes to pro-social behaviour. As well as categorising mechanisms of action according to these pre-existing theories, thought should be given to generating a more nuanced categorisation that distinguishes between different types of appeal to pro-social behaviour. 4) The document has excessive self-citation - more than one quarter of all citations. The world of applied behavioural science has substantially broader influences than this paper portrays and should be described as such. (Note that since I am myself already cited, this is not a self-interested comment, but a plea for acknowledgement of greater diversity of thought, influence and credit). Good luck with what is a very worthwhile project.
--

REVIEWER	Rebecca Webster University of Sheffield, UK
REVIEW RETURNED	26-Oct-2020

GENERAL COMMENTS	This is a very clear protocol detailing the procedure for a systematic review which will evaluate the effectiveness of prosocial
--

	public health messaging in encouraging behaviour relevant to reducing transmission of respiratory infections compared to messaging which focuses on self-protection. The results of this will help to design communication strategies that more effectively influence population behaviour during infectious disease outbreaks. The manuscript is clearly written, covering relevant and recent research leading to a well justified set of research questions and methods to answer these. I only have a few suggestions for the authors to consider.  1. In the introduction, when discussing the key principles for informing interventions to promote adherence to social distancing measure it may also be helpful for readers if you elaborate on principles 10, 11 and 12 as you have done for the previous 9. 2. As the introduction narrows from a broad overview of different communication strategies/key principles to then the key focus on prosocial public health messaging, I wonder if this should be better reflected in the title of the systematic review. At the moment 'The effect of communication strategies' could be referring to a variety of techniques so I would recommend referring to the exact strategy under investigation in the review. 3. Has the search strategy been piloted? For example, are you getting a feasible number of hits which include studies you are already aware of that would be eligible? I wonder if adding search terms such as 'flu'/'influenza' to section 1, 'messaging' to section 2, and 'compliance' to section 3 of your search strategy would return additional papers which could potentially be eligible. 4. For the primary outcomes extracted from included studies, what criteria are you using to separate outcomes in critical, important and not important? 5. It is now recommended that risk of bias in randomised trials is assessed using version 2 of the Cochrane Collaboration Risk of Bias Tool. See: https://methods.cochrane.org/bias/resources/rob-2-revised-cochrane-risk-bias-tool-randomized-trials
--	--

REVIEWER	David Rand MIT, USA
REVIEW RETURNED	27-Oct-2020

GENERAL COMMENTS	This is a timely proposal for a systematic review. It is well written and comprehensive. My main suggestion is to increase the scope of places to search for grey literature papers. In particular, the authors should also search Google Scholar, Psyarxiv (main location of psychology pre-prints), and SSRN (main location of economics pre-prints); as well as one of the various databases of published literature from economics as experimental economists are also working on this issue. One other question is whether the authors want to include the sharing of accurate vs inaccurate information about COVID-19 as a relevant behavior. I think this would be useful, given widespread concerns about COVID-19 misinformation and its ability to undermine other health-related behaviors. If the authors do decide to include interventions on information sharing, this paper of ours might be a useful starting place: Pennycook G, McPhetres J, Zhang Y, Lu JG, Rand DG. Fighting COVID-19 Misinformation on Social Media: Experimental Evidence for a Scalable Accuracy-Nudge Intervention. Psychological Science. 2020;31(7):770-780. doi:10.1177/0956797620939054
--

VERSION 1 – AUTHOR RESPONSE

Reviewer: 1	
1) An important limitation is omitted, which is the difficulty associated with trying to conduct a thorough review when so many studies are likely to be conducted but not written-up and peer-reviewed. Greater thought needs to be given to how this issue will be handled when weighting evidence.	Thank you for your comment. First we are looking at evidence broader than Covid-19 to learn from existing evidence. Second this is a general problem with all systematic reviews but we agree it is a particular problem where the evidence is developing rapidly. Nonetheless there are plenty of systematic reviews being done on Covid19-related topics and they are taking steps to be as up to date as possible by using pre-prints. While these have not been peer-reviewed we don't think this is a critical limitation for our review because we will review their quality independently. Indeed our review like most other reviews will admit studies that are published in the grey literature and so have not necessarily been through peer review.
2) I think the eligibility criteria need to be tweaked. A control group of 'no message' or 'self-protection' excludes control messages that contain no motivational content (e.g. purely informational messages). I think this exclusion would be a mistake.	Comparisons: Relevant comparisons will include no message or active control with messages focused on self-protection or messages that contain no motivational content (p. 7).
3) I have a concern that current behaviour change theories are at their weakest when it comes to pro-social behaviour. As well as categorising mechanisms of action according to these pre-existing theories, thought should be given to generating a more nuanced categorisation that distinguishes between different types of appeal to pro-social behaviour.	Thank you for your comment. While it is true that most psychological theories of health tend to focus on behaviour that is self-protective as opposed to protective of others, there are several theories that do focus on the latter such as psychology theories of social identity, social influence and moral behaviour. We will describe mechanisms of action including in terms of whether they offer an explicit and plausible means of promoting pro-social protective behaviours.
4) The document has excessive self-citation - more than one quarter of all citations. The world of applied behavioural science has substantially broader influences than this paper portrays and should be described as such. (Note that since I am myself already cited, this is not a self-interested comment, but a plea for acknowledgement of greater diversity of thought, influence and credit).	Thank you for bringing this to our attention. We deleted 4 self-citations and included 12 additional references for greater diversity (please see introduction and references sections).
Reviewer: 2	
1. In the introduction, when discussing the key principles for informing interventions to promote adherence to	Thank you for your comment. Please see introduction part (p. 5).

social distancing measure it may also be helpful for readers if you elaborate on principles 10, 11 and 12 as you have done for the previous 9.	
2. As the introduction narrows from a broad overview of different communication strategies/key principles to then the key focus on prosocial public health messaging, I wonder if this should be better reflected in the title of the systematic review. At the moment 'The effect of communication strategies' could be referring to a variety of techniques so I would recommend referring to the exact strategy under investigation in the review.	The title has been change as follows: "The effect of prosocial public health messages for population behaviour change in relation to respiratory infections: A Systematic Review protocol".
3. Has the search strategy been piloted? For example, are you getting a feasible number of hits which include studies you are already aware of that would be eligible? I wonder if adding search terms such as 'flu'/'influenza' to section 1, 'messaging' to section 2, and 'compliance' to section 3 of your search strategy would return additional papers which could potentially be eligible.	This was an initial/ indicative search strategy. We have designed a more robust search strategy with the professional support of an expert. We have included the new search strategy.
4. For the primary outcomes extracted from included studies, what criteria are you using to separate outcomes in critical, important and not important?	According to the Cochrane Collaboration recommendations, authors pre-specify the critical and important outcomes. We included only critical outcomes which is the behaviour relevant to reducing transmission of respiratory infections (e.g., socializing, hand hygiene, social distancing, face masks, touching face, catching coughs/sneezes, testing, reporting contacts, isolating if infected). Thus any other outcome reported in the study will be considered as not important (p. 9).
5. It is now recommended that risk of bias in randomised trials is assessed using version 2 of the Cochrane Collaboration Risk of Bias Tool. See: https://methods.cochrane.org/bias/resources/rob-2-revised-cochrane-risk-bias-tool-randomized-trials	Thank you for this suggestion. We replaced CCRBT with RoB 2 (p. 9).
Reviewer: 3	
1. My main suggestion is to increase the scope of places to search for grey literature papers. In particular, the authors should also search Google Scholar, Psyarxiv (main location of psychology pre-prints), and SSRN (main location of economics pre-prints); as well as one of the various databases of published literature from economics	For unpublished studies we will conduct searches in databases of "grey" literature such as PsycEXTRA, Social Science Research Network (SSRN), OSF PREPRINTS database which includes BioHackrXiv, Cogprints, MediArXiv, SocArXiv, P syArXiv and RePEc. We will also conduct supplementary searches in Google Scholar, hand search relevant journals, and conduct backward and forward citation searching of included studies and relevant reviews (p. 7).

as experimental economists are also working on this issue.	
2. One other question is whether the authors want to include the sharing of accurate vs inaccurate information about COVID-19 as a relevant behavior. I think this would be useful, given widespread concerns about COVID-19 misinformation and its ability to undermine other health-related behaviors. If the authors do decide to include interventions on information sharing, this paper of ours might be a useful starting place: Pennycook G, McPhetres J, Zhang Y, Lu JG, Rand DG. Fighting COVID-19 Misinformation on Social Media: Experimental Evidence for a Scalable Accuracy-Nudge Intervention. Psychological Science. 2020;31(7):770-780. doi:10.1177/0956797620939054	Thank you for your comment. We think this is a really interesting and important question but not one that can be rigorously examined in our review. We propose that our review should remain focused on the question of protecting others as this is a complex question on which we need to maintain a focus.

VERSION 2 – REVIEW

REVIEWER	Pete Lunn Behavioural Research Unit, ESRI, Ireland
REVIEW RETURNED	14-Dec-2020

GENERAL COMMENTS	The authors have addressed the concerns I raised adequately.
--

REVIEWER	Rebecca Webster University of Sheffield, UK
REVIEW RETURNED	26-Nov-2020

GENERAL COMMENTS	The authors have satisfactorily addressed my comments, I now recommend this paper is accepted for publication.
--

REVIEWER	David Rand MIT, USA
REVIEW RETURNED	19-Nov-2020

GENERAL COMMENTS	I look forward to seeing the results! Not at all necessary to include, but if it's helpful my coauthor and I have written a very concise set of guidelines for COVID-19 messaging that might be a useful complement to the guidelines described in the introduction: Yoeli, E., & Rand, D. G. (2020). A checklist for prosocial messaging campaigns such as COVID-19 prevention appeals. https://doi.org/10.31234/osf.io/rg2x9
--